# Effects of the COVID-19 pandemic on late postpartum women living with HIV in Kenya

**John M. Humphrey**[1]*, **Marsha Alera**[2], **Leslie A. Enane**[3], **Bett Kipchumba**[4],
**Suzanne Goodrich**[1], **Michael Scanlon**[5], **Julia Songok**[6], **Beverly Musick**[7],
**Lameck Diero**[8], **Constantin Yiannoutsos**[7], **Kara Wools-Kaloustian**[1]

**1** Division of Infectious Diseases, Department of Medicine, Indiana University School of Medicine,
Indianapolis, Indiana, United States of America, **2** Academic Model Providing Access to Healthcare
(AMPATH), Eldoret, Kenya, **3** The Ryan White Center for Pediatric Infectious Disease and Global Health,
Department of Pediatrics, Indiana University School of Medicine, Indianapolis, Indiana, United States of
America, **4** Department of Reproductive Health, Moi Teaching and Referral Hospital, Eldoret, Kenya,
**5** Indiana University Center for Global Health, Indianapolis, Indiana, United States of America, **6** Department
of Paediatrics, College of Health Sciences, Moi University, Eldoret, Kenya, **7** Department of Biostatistics and
Health Data Science, Indiana University, Indianapolis, Indiana, United States of America, **8** Department of
Medicine, College of Health Sciences, Moi University, Eldoret, Kenya

* humphrjm@iu.edu

## Abstract

Although an estimated 1.4 million women living with HIV (WHIV) are pregnant each year
globally, data describing the effects of the COVID-19 pandemic on postpartum women in
low- and middle-income countries (LMICs) are limited. To address this gap, we conducted
phone surveys among 170 WHIV ≥18 years and 18–24 months postpartum enrolled in HIV
care at the Academic Model Providing Access to Healthcare in western Kenya, and
assessed the effects of the pandemic across health, social and economic domains. We
found that 47% of WHIV experienced income loss and 71% experienced food insecurity dur-
ing the pandemic. The majority (96%) of women reported having adequate access to antire-
troviral treatment and only 3% reported difficulties refilling medications, suggesting that the
program's strategies to maintain HIV service delivery during the early phase of the pan-
demic were effective. However, 21% of WHIV screened positive for depression and 8% for
anxiety disorder, indicating the need for interventions to address the mental health needs of
this population. Given the scale and duration of the pandemic, HIV programs in LMICs
should work with governments and non-governmental organizations to provide targeted
support to WHIV at highest risk of food and income insecurity and their associated adverse
health outcomes.

Science, UNITED STATES

**Data Availability Statement:** All relevant data are
within the manuscript.

**Funding:** This research was supported by the
National Institute of Allergy and Infectious Diseases

## Introduction

The coronavirus disease 2019 (COVID-19) pandemic has disrupted economies and societies
worldwide [1]. It has also disrupted healthcare service delivery, including HIV service delivery
[2, 3]. Recent studies have shed light on the widescale impact of the COVID-19 pandemic on
different populations affected by HIV, finding that overall people living with HIV (PLHIV) are

(NIAID) and the Eunice Kennedy Shriver National Institute Of Child Health & Human Development (NICHD), in accordance with the regulatory requirements of the National Institutes of Health East Africa IeDEA Consortium (U01AI069911 to CY and KW). This work was also supported by the NICHD under Award Number K23HD105495 to JH. The content is solely the responsibility of the authors and does not necessarily represent the official views of the National Institutes of Health. The funders had no role in study design, data collection and analysis, decision to publish, or preparation of the manuscript.

**Competing interests:** The authors have declared that no competing interests exist.

at increased risk of adverse health outcomes due to COVID-19 and a myriad of adverse social and economic outcomes [4–6]. Pregnant and breastfeeding women have also been identified as a vulnerable population during the COVID-19 pandemic given the potential risk of adverse COVID-19 outcomes for this population, as well as depression, anxiety and other social and psychological issues during pregnancy and postpartum [7–9]. However, to date, studies of this population have largely focused on the perinatal period or within 6–12 months postpartum, despite little evidence that pandemic-related stressors, or women's vulnerability to them, decrease beyond these periods [10–14]. Given that an estimated 1.4 million women living with HIV (WHIV) are pregnant each year globally, understanding the challenges faced by this population in the late postpartum period (i.e. the period after 12 months postpartum) due to the COVID-19 pandemic is important and has not yet been described [15].

Characterizing the ways in which different sub-groups of PLHIV are affected by the COVID-19 pandemic is important, as they may have distinct vulnerabilities and needs. Such data are needed for HIV programs to effectively and efficiently implement interventions to support patients. Arbitrarily restricting interventions to women who are less than 12 months postpartum, for example, may exclude affected individuals beyond this period, unduly limiting the public health response. Like early postpartum women, late postpartum WHIV must also adapt to changes in autonomy and responsibilities as caregivers to young children while maintaining engagement in HIV care and adherence to antiretroviral treatment (ART). Late postpartum women may also remain vulnerable to COVID-19 control measures (e.g., 'lockdowns') that limit access to family, social and medical support at a time of acute need [16]. Quarantine, feelings of fatigue and isolation, fear that they or their children may become infected, and HIV- or COVID-related stigma are additional stressors that could affect late postpartum women's mental and psychological health, making it more difficult for them to manage other parental, household and work responsibilities, or adhere to ART [17, 18].

Kenya has one of the largest HIV burdens in the world with an HIV prevalence of 6.1% among women of childbearing age [15]. Knowledge about how the COVID-19 pandemic has impacted WHIV in countries with high HIV burdens is needed to support this population now and when future crises emerge. The objective of this study is to understand the social, economic and health impacts of the COVID-19 pandemic on late postpartum WHIV in Kenya. We hypothesized that the pandemic has affected late postpartum WHIV across multiple domains relating to their psychosocial wellbeing, social and economic capacity, and HIV care.

## Materials and methods

### Study setting

This study was conducted at clinical sites affiliated with the Academic Model Providing Access to Healthcare (AMPATH). AMPATH is a USAID-funded program that provides HIV services based on Kenyan HIV treatment guidelines to more than 200,000 patients at Ministry of Health clinics throughout western Kenya [19, 20]. AMPATH is the largest care program participating in the East Africa International Epidemiology Databases to Evaluate AIDS (EA-IeDEA) Consortium [21]. EA-IeDEA is one of seven regional data centers funded by the NIH to provide globally diverse HIV/AIDS data to streamline HIV/AIDS research.

Kenya reported its first case of COVID-19 in March 2020 [22]. The country has subsequently experienced five 'waves' of COVID-19 cases and deaths [23, 24]. The first wave occurred from June to September, and the second wave occurred from October to December, of 2020. Kenya experienced a third wave early in 2021, followed soon after by a fourth wave which began in June 2021. Most recently, from December 2021 to February 2022, Kenya

experienced its largest wave to date, driven primarily by the Omicron variant of SARS-CoV-2 which caused a similar surge in COVID-19 cases globally.

During each wave, the Government of Kenya implemented various control measures to reduce transmission of SARS-CoV-2, the virus causing COVID-19. These measures included restrictions on social gatherings and public transport, closures of schools, worship venues, bars and restaurants, curfews, border closures and cross-border and intercounty travel bans [25, 26]. The 2007–2008 post-election violence in western Kenya, which resulted in a large-scale humanitarian crisis, prompted similar governmental restrictions which caused disruptions in HIV care and ART adherence for children and adults living with HIV [27–29]. While the post-election violence was unanticipated and pre-planning to ensure consistent ART access was not undertaken, in the context of the COVID-19 pandemic, Kenya's Ministry of Health implemented strategies to ensure the continuation of HIV service delivery. These strategies were implemented at AMPATH-affiliated HIV clinics and included physical distancing in the HIV clinics, multi-month ART refills, screening of patients for COVID-19 symptoms, and scale-up of community-based ART distribution [30]. Despite these strategies, AMPATH-affiliated facilities have been acutely challenged by surges in patient care needs, limited care resources including personal protective equipment, and the deaths of several healthcare workers due to COVID-19 [31].

## Study population

From March 2018 to February 2019, EA-IeDEA conducted a prospective, observational study to determine the factors influencing retention in prevention of mother-to-child HIV transmission (PMTCT) care for pregnant and early postpartum WHIV who were retained in care and lost to follow-up (LTFU, defined as last clinic visit >90 days) [32, 33]. Eligibility criteria for WHIV in this parent study were: (i) age ≥18 years, (ii) pregnant or early (i.e. ≤6 months) postpartum, and (iii) enrollment at one of the following AMPATH-affiliated facilities during pregnancy: Busia District Hospital, Huruma Sub-District Hospital, Kitale District Hospital, Moi Teaching and Referral Hospital, and Uasin Gishu District Hospital. These hospitals are a mix of large and small facilities serving urban and rural populations and are representative of the various Ministry of Health facilities in the region. A convenience sample of retained WHIV were recruited by a research assistant at each facility to achieve a sample size proportional to the facility's antenatal clinic volume. LTFU WHIV from each facility were recruited through review of the facility registers and enrolled through phone contact and community tracking. The study was powered to detect a 10% difference in vertical transmission between infants of retained and LTFU WHIV, assuming 40% of women were LTFU.

To understand the range of effects these women have experienced during the COVID-19 pandemic, we recruited women previously enrolled in the EA-IeDEA PMTCT study to complete a phone-based survey (henceforth, 'COVID-19 survey') from August 1 to October 30, 2020, approximately 18–24 months post-delivery for all women enrolled in the parent study. We attempted to contact all women enrolled in the parent study to invite them to participate in the COVID-19 survey. At least three phone calls were made to contact each participant; women were excluded if the research team could not reach them after the third attempt.

## Data collection

The phone survey contained structured questions to capture the range of potential effects of the pandemic on WHIV within the following domains: COVID-19 knowledge and preparedness, social and economic impact of COVID-19, access and adherence to ART, health status and health seeking behavior, and mental health. The survey also contained the validated Patient Health Questionnare-2 (PHQ-2) depression and Generalized Anxiety Disorder-2

(GAD-2) screens, and an ART adherence questionnaire previously validated in western Kenya [34, 35]. The survey was designed to be implemented among different populations, including adolescents and non-pregnant adults, to facilitate future comparative analyses [36, 37]. Given the absence of baseline 'pre-pandemic' survey data, the survey questions were carefully worded to assess effects that could be potentially attributed to the pandemic (e.g., "During the outbreak, have you had greater difficulty than usual refilling your medications?").

Surveys were conducted in Kiswahili and English by research staff fluent in both languages. Participants' responses were collected and managed using REDCap electronic data capture tools hosted at Indiana University [38]. Participants whose responses indicated acute medical or psychological needs were referred to facility staff for additional support. The surveys also contained scripts instructing patients about how to protect themselves from COVID-19, including physical distancing, mask use, handwashing, self-isolating when symptomatic, and the importance of maintaining adherence to ART.

The survey data were linked to data collected at the time of the parent study for each participant, including: date of birth, educational attainment, marital status, employment, facility, number of children, date of delivery, timing of HIV diagnosis in relation to pregnancy, time on ART, and whether the participant had previously become LTFU from PMTCT care during their recent pregnancy or early postpartum.

### Ethics

This study was approved by the Moi University/Moi Teaching and Referral hospital Institutional Research and Ethics Committee in Kenya and the Indiana University Institutional Review Board in the United States. All participants provided verbal consent for participation in this COVID-19 sub-study.

### Analysis

We descriptively summarized the data according to each of the domains of the survey, using frequencies and proportions for categorical variables and medians and interquartile ranges for continuous variables. PHQ-2 scores $\geq 3$ were considered a positive screen requiring referral for evaluation of possible depression [34]. GAD-2 scores $\geq 3$ were considered a positive screen for elevated anxiety symptoms warranting clinical referral [39]. The chi-square test was used to assess associations between subgroups including those enrolled in the COVID-19 study who were retained or LTFU at enrollment in the parent study, respectively, as well as between those who did not have public-facing work prior to the pandemic and had positive mental health screening or food insecurity during the pandemic, respectively. SAS software was used for the analysis [40].

### Results

Among 334 WHIV enrolled in the parent study, 164 (49%) were not enrolled in the COVID-19 study for the following reasons: 113 (69%) did not answer the phone after $\geq 3$ attempts or had nonworking numbers, 41 (25%) had no contact information in their medical records, 4 (2.4%) could not be contacted because a non-participant answered the phone and would not refer the potential participant to be contacted by study team, and 2 (1.2%) had died since the time they enrolled in the parent study. Of the 174 women who were contacted to participate in the COVID-19 study, 4 (2.3%) declined consent, leaving a total of 170 women (51% of the parent study cohort) who completed the survey and were included in the analysis.

At the time women were enrolled in the parent study, their median (IQR) age was 32 (27–35) years, 51% had completed secondary or tertiary school, 79% were married or cohabitating,

and 28% were newly diagnosed with HIV during their recent pregnancy (Table 1). A higher proportion of women who were retained in care at enrollment in the parent study completed the COVID-19 survey compared to women who were LTFU at enrollment in the parent study: 133 of 235 (57%) previously retained women were enrolled in the COVID-19 study, compared to 37 of 99 (37%) previously LTFU women (*P* = 0.001).

## COVID-19 knowledge and preparedness

Nearly all late postpartum WHIV (99%) had heard of COVID-19, with the most common sources of information consisting of TV news or radio ($\geq$90% each), followed by family, friends and text messages (approximately 60% each), and WhatsApp and Facebook (approximately 30% each) (Table 2). Twenty-three percent of women reported having received information about COVID-19 from their healthcare provider or clinic. When asked to describe the symptoms of COVID-19, the most common responses were cough, fever, and shortness of breath ($\geq$80% each). Loss of taste or smell was reported by 19% of women. When asked what steps should be taken if one has symptoms of COVID-19, >90% of women replied that one should contact and/or visit a health facility and isolate from others, respectively. None of the women suggested visiting a traditional/spiritual healer or herbalist.

Concerning late postpartum women's preparedness for COVID-19, 15% were concerned about staying healthy during the pandemic, and 95% perceived that all types of people were

**Table 1. Characteristics of WHIV at enrollment in the parent study.**

| | Total N = 170 n (%) |
|---|---|
| **Characteristics** | |
| Age, median years (IQR)[a] | 32 (27–35) |
| Educational attainment | |
| None | 14 (8.2) |
| Primary school | 69 (41) |
| Secondary school | 62 (36) |
| Tertiary school | 25 (15) |
| Marital status | |
| Single | 29 (17) |
| Married or cohabitating | 135 (79) |
| Separated or divorced | 4 (2.4) |
| Widowed | 2 (1.2) |
| Facility | |
| Busia District Hospital | 29 (17) |
| Huruma Sub-District Hospital | 25 (15) |
| Kitale District Hospital | 18 (11) |
| Moi Teaching and Referrral Hospital | 69 (41) |
| Uasin Gishu District Hospital | 29 (17) |
| Number of children, median (IQR)[b] | 2 (1–3) |
| Newly diagnosed with HIV during recent pregnancy | 47 (28) |
| Months on ART[c] | 27 (4.8–76) |
| Previously disengaged from care | 37 (78) |

[a] n = 164

[b] n = 133 women with other children

[c] n = 168

**Table 2. COVID-19 knowledge and preparedness among late postpartum WHIV.**

| | Total N = 170 n (%) |
|---|---|
| **COVID-19 Knowledge** | |
| Heard of COVID-19 | 169 (99) |
| Main sources of information about COVID-19[a] | |
| TV news | 156 (92) |
| Radio | 152 (90) |
| Friends | 115 (68) |
| Family | 113 (67) |
| Text messages | 105 (62) |
| WhatsApp | 65 (39) |
| Facebook | 57 (34) |
| Healthcare provider or clinic | 38 (23) |
| Internet news site | 9 (5.3) |
| Received official communication from HIV clinic about COVID[a] | 68 (40) |
| Advice on how to stay healthy | 64 (94) |
| Suggestions to keep supply of ART | 14 (21) |
| Advice on what to do if feeling sick | 12 (18) |
| Symptoms independently identified for COVID[a] | |
| Cough | 158 (94) |
| Fever | 157 (93) |
| Shortness of breath | 138 (82) |
| Headache | 61 (36) |
| Loss of taste or smell | 32 (19) |
| Body aches | 30 (18) |
| Don't know | 6 (3.6) |
| Vomiting | 3 (1.8) |
| Diarrhea | 1 (0.6) |
| Actions to take if sick with symptoms of COVID | |
| Contact and/or visit health facility | 160 (95) |
| Isolate from others | 156 (92) |
| Manage symptoms from home if able | 7 (4.1) |
| Visit traditional/spiritual healer or herbalist | 0 (0) |
| Don't know | 8 (4.7) |
| **COVID-19 Preparedness** | |
| Concerned about being able to stay healthy during pandemic | 26 (15) |
| Perception that all types of people can get coronavirus | 162 (95) |
| Precautions taken to protect self from coronavirus | |
| Washing hands frequently | 169 (99) |
| Covering face with a mask or scarf | 157 (92) |
| Avoiding visitors or social gatherings | 152 (89) |
| Avoiding touching face | 92 (54) |
| Avoiding people who are sick | 19 (11) |
| Not going to church | 3 (1.8) |
| Avoiding public transportation | 2 (1.2) |
| Not going to work site or not going to do informal work | 0 (0) |
| Reasons for having to leave home | |
| Getting food our household supplies | 166 (98) |
| Picking up medicine or accessing medical care | 160 (94) |

(*Continued*)

**Table 2.** (Continued)

| | Total N = 170 n (%) |
|---|---|
| Going to work site | 121 (71) |
| Going to church | 83 (49) |
| Caring for others | 18 (11) |
| Have not left home | 0 (0) |

[a] n = 169

susceptible to COVID-19 (Table 2). When women were asked what measures they practiced to protect themselves from COVID-19, ≥80% reported washing hands frequently, covering their face with a mask or scarf, and avoiding visitors or social gatherings, respectively. Few women (1%) reported avoiding social environments such as church and public transport. All women reported having left home since the start of the pandemic, most commonly to get food or household supplies (98%), pick up medicine or access care (94%), and go to work (71%).

## Social and economic impact of COVID-19

Two thirds of late postpartum women reported living in their usual residence during the pandemic, while a third reported living at their rural home (Table 3). Most women (75%) had a

**Table 3. Social and economic impacts of the COVID-19 pandemic among late postpartum WHIV.**

| | Total N = 170 n (%) |
|---|---|
| Living situation during pandemic | |
| Usual residence | 111 (65) |
| Rural home | 52 (31) |
| Work site | 4 (2.3) |
| Relative's home | 3 (1.8) |
| Need to travel from town to rural area because of pandemic | 44 (26) |
| History of working or income generation prior to pandemic | 128 (75) |
| Formal or informal sector | |
| Formal sector | 16 (12) |
| Informal sector | 112 (88) |
| Public-facing work[a] | 121 (95) |
| Loss of job/income during pandemic[a] | |
| No loss of job or income | 61 (48) |
| Loss of other income | 60 (47) |
| Loss of formal job | 7 (5.5) |
| Others at home relying on woman's income[a] | 87 (68) |
| Others at home (whom the woman relies on for their income) have lost job or lost income during the pandemic | 90 (53) |
| Food insecurity during pandemic | 121 (71) |
| Sources of food during pandemic | |
| Stores or markets | 167 (98) |
| Own farm | 72 (42) |
| Relative's or friend's farm | 61 (36) |

[a] percentages among 128 women with history of working or income generation prior to pandemic.

history of working or income generation prior to the COVID-19 pandemic, of which 88% was in the informal sector and 95% was public facing. Nearly half of women (47%) reported loss of income during the pandemic, and 53% of all women reported that others at home had lost their job or income during the pandemic. In addition, 71% of women reported experiencing food insecurity during the pandemic.

### Access and adherence to ART

During the pandemic, 96% of late postpartum women reported that they had access to ART at the time of the survey (Table 4). A total of 6.5% were concerned about running out of ART, and 7.1% reported having skipped ART doses due to concerns about running out of ART. Overall, 3.6% of women reported greater difficulties than usual refilling medications during the pandemic, most commonly due to the cost or lack of availability of transportation. Only 3% of women reported problems taking ART, 6% reported forgetting to take ART in the past 3 days, and 3.6% reported missing ≥1 ART dose in the past 3 days. However, 19% of women reported having to make changes in the way they take ART because of the pandemic, such as

**Table 4. ART access and adherence among late postpartum WHIV during the COVID-19 pandemic.**

|  | Total N = 170 n (%) |
|---|---|
| **ART Access** |  |
| Currently has ART[a] | 162 (96) |
| ART quantity remaining[b] |  |
| ≤1 week | 4 (2.5) |
| >1 week and ≤30 days | 30 (19) |
| >30 days and ≤60 days | 68 (42) |
| >60 days and ≤90 days | 55 (34) |
| >90 days | 5 (3.1) |
| Concerned about running out of ART[a] | 11 (6.5) |
| Skipped ART doses during pandemic due to concerns about running out[a] | 12 (7.1) |
| Had greater difficulties than usual refilling medications during pandemic[a] | 6 (3.6) |
| Transportation too expensive or not available | 5 (83) |
| Curfew, restrictions on travel, or police crackdowns | 1 (17) |
| Clinic or facility not open, not functioning, or stock-outs | 0 (0) |
| Fear of traveling | 0 (0) |
| Has been sick | 0 (0) |
| Household or family members unwell | 0 (0) |
| Other | 1 (17) |
| **ART Adherence[c]** |  |
| Problems taking ART | 5 (3.0) |
| Forgot to take ART in past 3 days | 10 (6.0) |
| Problems taking ART around others | 9 (5.4) |
| Took an ART dose >1 hour late in past 7 days | 45 (27) |
| Times when supposed to take ART but did not have it with her | 11 (6.6) |
| Missed ≥1 ART dose in past 7 days | 6 (3.6) |
| Made changes in taking ART because of pandemic | 31 (19) |

N/A, not applicable

[a] n = 169

[b] percentages among 162 women with ART remaining

[c] percentages in this category are among 166 women self-reported to be on ART

**Table 5. Health status and health-seeking behavior among late postpartum WHIV during the COVID-19 pandemic.**

|  | Total N = 170 n (%) |
| --- | --- |
| Felt ill during past 2 months[a] | 30 (18) |
| Any symptom during past 2 months | 64 (38) |
| Sought care[b] | 22 (73) |
| Hospitalized during past 2 months | 1 (0.6) |
| Location of care[c] |  |
| Clinic | 21 (96) |
| Hospital | 0 (0) |
| Pharmacist | 0 (0) |
| Household members or other close contacts have been ill | 8 (4.7) |
| Had COVID-19 laboratory test[a] | 13 (7.7) |
| Diagnosed with COVID[a] | 15 (8.9) |
| Having greater difficulties than usual in getting medical care | 2 (1.2) |

[a] n = 169; frequency of reported symptoms during the past 2 months include congestion (n = 17), body aches (n = 15), cough (n = 14), fever (n = 10), fatigue (n = 5), vomiting (n = 3), shortness of breath (n = 3), sore throat (n = 2), diarrhea (n = 2), other (n = 16)

[b] percentages among 30 women who felt ill during past 2 months

[c] percentages among 22 women who sought care

adjusting the timing of ART dosages to accommodate work and family schedule changes, or to coincide with the availability of food.

## Health status and health seeking behavior

Eighteen percent of late postpartum women reported feeling ill during the past two months, and of these 70% sought care at a clinic and one was hospitalized (Table 5). However, when women were asked whether they had experienced any COVID-related symptoms over the past two months, 38% reported experiencing at least one symptom. A total of 7.7% of women reported having had a COVID-19 test, and 8.9% reported being diagnosed with COVID-19 by a medical provider (either based on symptoms or a laboratory test). Only 1.2% of women reported greater difficulties than usual getting medical care during the pandemic.

## Mental health

PHQ-2 scores were ≥3 in 21% of participants, indicating a positive screen for possible depression. GAD-2 scores were ≥3 in 8% of participants, indicating a positive screen for elevated anxiety symptoms. Not having public-facing work prior to the pandemic was not associated with a positive PHQ-2 or GAD-2 screen or experiencing food insecurity during the pandemic, respectively ($p > 0.05$ for all).

## Discussion

In this study, we describe the impact of the COVID-19 pandemic on late postpartum WHIV in western Kenya. Almost half of WHIV experienced income loss and a majority experienced food insecurity during the pandemic. In addition, nearly all women reported having adequate access to antiretroviral treatment and only 3% reported difficulties refilling medications, suggesting that the HIV program's strategies to maintain HIV service delivery during the early phase of the pandemic were effective. Still, one in five late postpartum WHIV screened positive

for depression and 8% for anxiety disorder, indicating the need for interventions to address the mental health needs of this population.

We also found that COVID-19 knowledge was high among late postpartum WHIV in our study, and that >90% reported that traditional media, including TV news and radio, were their main sources of COVID-19 information. In contrast, only a third of late postpartum WHIV cited that the social media platforms WhatsApp and Facebook comprised their main sources of COVID-19 information. In recent years, policymakers and researchers have promoted the role of social media as an intervention to enhance HIV prevention and treatment efforts in LMICs [41]. This includes leveraging social media to enhance communication between patients and providers, disseminate information, and develop support networks. However, social media was not a primary source of COVID-19 information for most WHIV in our study. This finding carries important implications for the potential reach of social media-based healthcare interventions for this population in the future. Furthermore, only 23% of late postpartum WHIV received COVID-19 information from their provider or clinic, despite the vast majority stating that the pandemic had not disrupted their access to care. These findings suggest that multipronged communication strategies are likely needed to effectively communicate public health messages to WHIV in this setting.

Half of late postpartum WHIV in our study experienced job or income loss during the pandemic, and 71% experienced food insecurity. These alarming findings indicate that this group, whose income is predominantly from the informal sector, are deeply vulnerable to the social and economic stressors precipitated by the pandemic. These findings also align with other studies that describe the ways in which the COVID-19 pandemic has exacerbated food insecurity in East Africa [42, 43]. Food insecurity is highly prevalent in western Kenya and the surrounding region, occurring in more than 50% of adults irrespective of HIV status, and may have been exacerbated by the COVID-19 pandemic [44, 45]. For PLHIV, food insecurity has been shown to be bidirectionally linked to a myriad of nutritional, behavioral, economic, and health outcomes, including HIV treatment outcomes [45, 46]. As such, this finding could also represent an early warning signal of the pandemic's effect on future health-related outcomes for WHIV as well as additional effects not captured in our survey. AMPATH has implemented various interventions to enhance food and economic security in western Kenya over the years, including direct food distribution and microfinance groups [47–49]. However, donor-driven HIV programs like AMPATH will not have the capacity to acutely scale and sustain interventions that address food and economic insecurity at the population level during the pandemic, and thus will need actionable strategies in collaboration with governmental and non-governmental organizations (NGOs) to provide resources targeted to those at highest risk of adverse health outcomes.

Despite the heavy burden of economic and food insecurity reported by late postpartum WHIV in our study, very few women reported that the pandemic had adversely impacted their ability to access care and adhere to ART. This is significant considering prior literature correlating social and economic stressors, including food insecurity, with ART non-adherence [50, 51]. This finding may reflect the characteristics of the women in our sample, who were predominantly retained in care and were not newly diagnosed with HIV during their recent pregnancy. Still, these findings suggest that these WHIV have successfully adapted to the challenges posed by the early phase of the COVID-19 pandemic. These findings may also be driven in part by adaptations made by the AMPATH program during the pandemic, such as multi-month refills, less frequent appointments, and other differentiated service delivery strategies. More data are needed to understand how ART outcomes are affected as the pandemic evolves. Programs will probably need to mobilize proven strategies to maintain patients' access to care and ART adherence, such as peer counsellors, SMS communications, cash transfers and other measures [52].

Finally, 21% of late postpartum WHIV screened positive for depression and 8% for generalized anxiety disorder based on the PHQ-2 and GAD-2, respectively. These outcomes may be intertwined with the high prevalence of economic and food insecurity observed in our study, as well as reduced social support experienced during the pandemic and HIV-related stigma [53]. Prior studies in Kenya have reported postpartum depression rates of 13–18% among HIV uninfected women and 48% among WHIV [34, 53, 54]. Thus, the degree to which our findings reflect baseline screening rates, versus those attributable to the pandemic, is unclear. It is also important to note that these screening tools are designed to prompt further evaluation for mental health disorders rather than formal diagnoses, so more data will be needed to define the true burden of mental illness among late postpartum WHIV. Nevertheless, these findings should alert HIV programs to the need to further define and address the burden of mental illness among late postpartum WHIV during the COVID-19 pandemic.

A strength of our study is its diverse sample of late postpartum WHIV from a variety of urban and peri-urban health facilities in western Kenya. Our inclusion of women who were previously LTFU is also a strength, as these individuals may have key vulnerabilities which drove their prior disengagement from care, and which were exacerbated by the COVID-19 pandemic. Selection bias likely limits the generalizability of our findings. A high proportion of women enrolled in our parent study could not be re-contacted due to missing or nonviable contact information in their medical records, and recruitment was significantly lower among previously LTFU compared to retained WHIV. This important observation suggests that the challenges faced by WHIV are probably underreported in our study, and that enhanced efforts will be needed for programs to reach their most vulnerable patients and meet their needs. Our data are also subject to social desirability and other reporting biases, which may have resulted in over-reporting of ART adherence and the under-reporting of mental health-related symptoms. Finally, although our study provides a rich snapshot of the pandemic's impact on late postpartum WHIV in Kenya, programs in Kenya and other LMICs will also need to examine the pandemic's longitudinal effects on WHIV as it evolves, as well as more deeply explore the interplay between womens' postpartum experiences and the challenges of the COVID-19 pandemic.

## Conclusion

Food insecurity and loss of economic capacity were highly prevalent among late postpartum WHIV during the COVID-19 pandemic in western Kenya. Given the scale and duration of the pandemic, HIV programs in LMICs should work with governments and NGOs to provide targeted support to the patients at highest risk of food and income insecurity and the adverse health outcomes associated with them. Further research is needed to explore the longitudinal impact of the pandemic on this population, and the mechanisms mediating the effects of the pandemic across the spectrum of health, social and economic outcomes for late postpartum WHIV.

## Supporting information

**S1 Questionnaire. Inclusivity in global research.**
(DOCX)

## Acknowledgments

We thank the women who participated in this study. We also thank Justin Kipsang and the clinical staff at Moi Teaching and Referral Hospital for collecting the data that made this analysis possible.

## Author Contributions

**Conceptualization:** John M. Humphrey, Leslie A. Enane, Suzanne Goodrich, Kara Wools-Kaloustian.

**Data curation:** Marsha Alera, Michael Scanlon, Beverly Musick.

**Formal analysis:** Beverly Musick, Constantin Yiannoutsos, Kara Wools-Kaloustian.

**Funding acquisition:** Lameck Diero, Constantin Yiannoutsos, Kara Wools-Kaloustian.

**Methodology:** Leslie A. Enane, Bett Kipchumba, Suzanne Goodrich, Beverly Musick, Constantin Yiannoutsos, Kara Wools-Kaloustian.

**Project administration:** Marsha Alera, Bett Kipchumba, Michael Scanlon.

**Resources:** Lameck Diero, Constantin Yiannoutsos, Kara Wools-Kaloustian.

**Supervision:** Michael Scanlon, Julia Songok, Lameck Diero, Constantin Yiannoutsos, Kara Wools-Kaloustian.

**Writing – original draft:** John M. Humphrey.

**Writing – review & editing:** John M. Humphrey, Marsha Alera, Leslie A. Enane, Bett Kipchumba, Suzanne Goodrich, Michael Scanlon, Julia Songok, Beverly Musick, Constantin Yiannoutsos, Kara Wools-Kaloustian.

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
