## [Decision Letter · Decision Letter 0]

4 Oct 2022

PGPH-D-22-00598

Effects of the COVID-19 pandemic on late postpartum women living with HIV in Kenya

Dear Dr. Humphrey,

Thank you for submitting your manuscript to PLOS Global Public Health. After careful consideration, we feel that it has merit but does not fully meet PLOS Global Public Health’s publication criteria as it currently stands. Therefore, we invite you to submit a revised version of the manuscript that addresses the points raised during the review process.

We look forward to receiving your revised manuscript.

Kind regards,

Ahmed Waqas

Academic Editor

Journal Requirements:

2. In the online submission form, you indicated that "Data underlying the reported findings in this study are available upon request made to the corresponding author." All PLOS journals now require all data underlying the findings described in their manuscript to be freely available to other researchers, either 1. In a public repository, 2. Within the manuscript itself, or 3. Uploaded as supplementary information.

Additional Editor Comments (if provided):

Reviewers' comments:

Reviewer's Responses to Questions

**Comments to the Author**

1. Does this manuscript meet PLOS Global Public Health’s publication criteria? Is the manuscript technically sound, and do the data support the conclusions? The manuscript must describe methodologically and ethically rigorous research with conclusions that are appropriately drawn based on the data presented.

Reviewer #1: Yes

Reviewer #2: No

Reviewer #3: Yes

2. Has the statistical analysis been performed appropriately and rigorously?

Reviewer #1: Yes

Reviewer #2: Yes

Reviewer #3: Yes

3. Have the authors made all data underlying the findings in their manuscript fully available (please refer to the Data Availability Statement at the start of the manuscript PDF file)?

Reviewer #1: Yes

Reviewer #2: Yes

Reviewer #3: Yes

4. Is the manuscript presented in an intelligible fashion and written in standard English?

Reviewer #1: Yes

Reviewer #2: Yes

Reviewer #3: Yes

5. Review Comments to the Author

Reviewer #1: Overall, this manuscript was well-written and provides useful information to inform practice, policy, and future research focused on supporting pregnancy and postpartum WHIV. I would overall recommend providing a stronger argument and clarifying the reason for the focus on the late-postpartum period and to also be more specific to this population in the discussion. Specific feedback is below.

Abstract – well-written

Introduction –

1. This sentence is awkward: “Pregnant and breastfeeding women have also been identified as a vulnerable population during the COVID-19 pandemic given the potential risk of COVID-19 on maternal and newborn outcomes, as well as depression, anxiety and other social and psychological issues during pregnancy and postpartum.” Would recommend editing to read: “Pregnant and breastfeeding women have also been identified as a vulnerable population during the COVID-19 pandemic given the potential risk of adverse COVID-19 outcomes for this population, as well as….”

2. “Yet although an estimated 1.4 million women living with HIV (WHIV) are pregnant each year globally, the unique challenges faced by this population in the late postpartum period (i.e. the period after 18 months postpartum) due to the COVID-19 pandemic have not yet been described [10].” – what is the evidence that exists re: the unique challenges of the WHIV population due to COVID-19 in pregnancy and early postpartum? You have, up to this point, only discussed general concerns of the pregnant and postpartum population due to COVID-19 – need to fill in this gap before jumping to the need to discuss late-postpartum.

3. Second paragraph – still not clear why the late postpartum period is more vulnerable to some of the listed concerns than at other times in pregnancy. Why, in particular, is this time period the focus?

4. Overall, need to strengthen argument/reasoning for targeting late-postpartum period…seem to switch between discussing affects on WHIV overall and then those who are pregnant – more consistency is needed.

Methods/Materials

1. Please define PMTCT

2. Please more clearly define eligibility for this study – after reading the “study population” section several times, it became clear that the eligibility described in the first paragraph of that section was for a larger study and that the population for this study was recruited from there. Please define more clearly who “them” is when stating you “recruited them” and how it is separate from the parent study (also, recommended to reference the parent study if results are published).

3. Why were there not questions specifically related to their postpartum experience, since the target population was late-postpartum (i.e. things such as a difficult birth, morbidities postpartum, etc…could greatly impact their ability to access care, work, etc…)?

Analysis

- Some more detail is needed here as based on sharing a p-value in the results, certain comparative analyses, such as chi square and or t-tests, were used.

Results

- Could be helpful to understand if there was any difference in concerns such as food insecurity and even positive mental health screenings between those who have work outside the home and those who didn’t prior to the pandemic, particularly knowing the association between depression and anxiety and the social distancing implemented during the pandemic.

-

Discussion

- Would be helpful for the first paragraph to focus on providing an overall summary of the results, then move into connecting it to current literature.

- Limitations well-described

- As stated in with regard to the introduction – the population of focus was WHIV in the late postpartum period, yet the discussion often focuses on WHIV overall. This type of generalization needs to be justified or more specific wording is needed to clarify you are meaning WHIV in the postpartum period.

- Per the above, consider phrasing the last sentence of the conclusion to state “spectrum of health, social and economic outcomes for late postpartum WHIV.

Reviewer #2: In this study, Humphrey et al., have attempted to understand the social, economic and health impacts of the COVID-19 pandemic on late postpartum WHIV in Kenya. This is a question of public health importance and the authors must be commended for this.

I have some concerns on the validity of the estimates though.

1. Only 47% of all the contacted women have responded and participated in the study. This is a very sub-optimal response rate. The tendency to respond to the survey has the potential to bias the estimates in one way or the other and the most disturbing concern is that we would never know the extent and direction of such a bias.

2. Without the lack of pre-pandemic data, the burden of mental illnesses such as depression and generalized anxiety disorder attributed to the pandemic is difficult to assess.

With these concerns, I’m afraid that I’m less inclined to reconcile with the estimates reported in the manuscript.

Reviewer #3: The manuscript is technically sound and the data does support the conclusions of the study. The authors do adequately highlight the many limitations of the study and non generalizability of the results. The statistical analysis been performed appropriately and rigorously though a few areas need clarity as indicated in the additional comments below. All data underlying the findings in the manuscript is fully available. The manuscript is generally presented in an intelligible fashion and written in standard English

Additional comments

1.More detail is required by the authors on the sample size calculation for the parent study. Given the scale of the AMPATH program, how was the sample size determined? The exclusion criteria for the parent study is not indicated. These details are important as they have a direct bearing on the COVID 19 survey results.

2. Under 'access and adherence to ART' section of the results, it is not clear what changes the 19% of women had to make in the way they take ART due to the pandemic. The authors need to explain what changes these were.

6. PLOS authors have the option to publish the peer review history of their article (what does this mean?). If published, this will include your full peer review and any attached files.

**Do you want your identity to be public for this peer review?** For information about this choice, including consent withdrawal, please see our Privacy Policy.

Reviewer #1: No

Reviewer #2: **Yes: **Ramachandran Thiruvengadam

Reviewer #3: **Yes: **Morrison Zulu

---

## [Decision Letter · Decision Letter 1]

27 Feb 2023

PGPH-D-22-00598R1

Effects of the COVID-19 pandemic on late postpartum women living with HIV in Kenya

Dear Dr. Humphrey,

Thank you for submitting your manuscript to PLOS Global Public Health. After careful consideration, we feel that it has merit but does not fully meet PLOS Global Public Health’s publication criteria as it currently stands. Therefore, we invite you to submit a revised version of the manuscript that addresses the points raised during the review process.

Overall, the reviewers felt that their previous comments were addressed appropriately. However, reviewer #1 raised two minor points regarding the interpretation and discussion of the results, which should be addressed. 

We look forward to receiving your revised manuscript.

Kind regards,

Alex Schaefer, PhD

Associate Editor

Journal Requirements:

2. We have noticed that you have uploaded Supporting Information files, but you have not included a list of legends. Please add a full list of legends for your Supporting Information files after the references list. 

Additional Editor Comments (if provided):

Reviewers' comments:

Reviewer's Responses to Questions

**Comments to the Author**

1. If the authors have adequately addressed your comments raised in a previous round of review and you feel that this manuscript is now acceptable for publication, you may indicate that here to bypass the “Comments to the Author” section, enter your conflict of interest statement in the “Confidential to Editor” section, and submit your "Accept" recommendation.

Reviewer #1: (No Response)

Reviewer #3: All comments have been addressed

2. Does this manuscript meet PLOS Global Public Health’s publication criteria? Is the manuscript technically sound, and do the data support the conclusions? The manuscript must describe methodologically and ethically rigorous research with conclusions that are appropriately drawn based on the data presented.

Reviewer #1: Yes

Reviewer #3: Yes

3. Has the statistical analysis been performed appropriately and rigorously?

Reviewer #1: Yes

Reviewer #3: Yes

4. Have the authors made all data underlying the findings in their manuscript fully available (please refer to the Data Availability Statement at the start of the manuscript PDF file)?

Reviewer #1: No

Reviewer #3: Yes

5. Is the manuscript presented in an intelligible fashion and written in standard English?

Reviewer #1: Yes

Reviewer #3: Yes

6. Review Comments to the Author

Reviewer #1: Overall, comments were addressed well. Two minor comments:

1. The comment regarding examining associations between having or not having a public facing job and positive PHQ-2 or GAD-2 wasn't answered quite right (comment #9), although I don't think it's essential to include this analysis. My comment stated that it would be interesting to look at associations between those who had public facing jobs versus those who did not with respect to food insecurity and/or GAD-2 and PHQ-2. i.e. was there an association between those who did NOT have a public facing job and a positive GAD-2 or PHQ-2? I would expect there might be - due to potentially more acute social isolation experienced by those who did not work. I also would think food insecurity might be more likely for that group.

2. The first paragraph of the results could be a more general summary - i.e. instead of repeating results, what is the overall summary you want to leave the readers with? .i.e. the second sentence could say "Almost half of WHIV experienced income loss and a majority experienced food insecurity during the pandemic."

Reviewer #3: Previous reviewer comments have been addressed by the authors

7. PLOS authors have the option to publish the peer review history of their article (what does this mean?). If published, this will include your full peer review and any attached files.

**Do you want your identity to be public for this peer review?** For information about this choice, including consent withdrawal, please see our Privacy Policy.

Reviewer #1: No

Reviewer #3: **Yes: **Morrison Zulu

---

## [Editor Report · Decision Letter 2]

10 Mar 2023

Effects of the COVID-19 pandemic on late postpartum women living with HIV in Kenya

PGPH-D-22-00598R2

Dear Dr. Humphrey,

We are pleased to inform you that your manuscript 'Effects of the COVID-19 pandemic on late postpartum women living with HIV in Kenya' has been provisionally accepted for publication in PLOS Global Public Health.

Best regards,

Julia Robinson

Executive Editor